# Spectral Object Recognition in Hyperspectral Holography with Complex-Domain Denoising

**DOI:** 10.3390/s19235188

**Published:** 2019-11-26

**Authors:** Igor Shevkunov, Vladimir Katkovnik, Daniel Claus, Giancarlo Pedrini, Nikolay V. Petrov, Karen Egiazarian

**Affiliations:** 1Faculty of Information Technology and Communication Sciences, Tampere University, FI-33101 Tampere, Finland; vladimir.katkovnik@tuni.fi (V.K.); karen.eguiazarian@tuni.fi (K.E.); 2Department of Photonics and Optical Information Technology, ITMO University, 197101 St. Petersburg, Russia; n.petrov@niuitmo.ru; 3Institut für Lasertechnologien in der Medizin und Messtechnik, Helmholtzstraße 12, 89081 Ulm, Germany; daniel.claus@ilm-ulm.de; 4Institut für Technische Optik (ITO), Universität Stuttgart, Pfaffenwaldring 9, 70569 Stuttgart, Germany; pedrini@ito.uni-stuttgart.de

**Keywords:** hyperspectral imaging, singular value decomposition, sparse representation, noise filtering, noise in imaging systems

## Abstract

In this paper, we have applied a recently developed complex-domain hyperspectral denoiser for the object recognition task, which is performed by the correlation analysis of investigated objects’ spectra with the fingerprint spectra from the same object. Extensive experiments carried out on noisy data from digital hyperspectral holography demonstrate a significant enhancement of the recognition accuracy of signals masked by noise, when the advanced noise suppression is applied.

## 1. Introduction

Hyperspectral imaging (HSI) is used to obtain the spectral distributions for each pixel of the image of a scene. One of the main goals of this spectral analysis is to classify an image and its fragments. This is possible due to availability of the detailed spectral data with thousands of spectrum lines typical for HSI. The specific spectra, called fingerprints or spectrum signatures, are used for object recognition and classification.

HSI is extremely effective in obtaining spectral data in many applications such as earth remote sensing [1], terahertz imaging [2], and medical imaging [3]. For example, highly spectrally resolved images provide spectral signatures unique to facial skin tissue that may not be detected using common CCD cameras, which cannot distinguish well between different spectral signals [4]. Spectral recognition is well known in Fourier transform infrared spectroscopy (FTIR), which provides a biochemical fingerprint of a biopsy sample and, together with advanced data analysis techniques, can classify cells. However, one of the challenges when dealing with FTIR imaging is the slow recording of the data such that a one square centimeter tissue section requires several hours of image recording [5].

Hyperspectral Digital Holography (HSDH) is a technique based on FT spectrometer but uses multipixel detectors instead of single-pixel ones like in FTIR, with that advantage scanning of tissues is possible $10^{6}$ faster just due to the larger area of the detector. As an additional advantage, HSDH provides information on both object amplitude and phase in each image pixel [6,7]. With that information, one can obtain spectra from the area of interest and provide a recognition and/or process detection. However, HS data are severely corrupted by noise, which drastically changes the shape of spectra and ruins the recognition process. It is a major challenge that analytical biosensing platforms face, as the signal from ensemble measurements, at low concentrations, is masked by the background noise [8].

Traditionally, many rooting techniques are used for denoising in HSDH, but they do not help much in noise suppression, working for low noise levels [7,9], but failing in the high ones [10]. It is explained by a slice-wise separate filtering which does not process all HS cube slices jointly. In particular, they fail due to high noise levels on spectral components with low intensity of a radiation source, which causes a low signal-to-noise ratio (SNR). An additional challenge in HSDH is a phase filtering, which can not be done by a traditional image noise suppression due to the angle-like behavior of the phase. Recently, a noise suppression technique for complex-domain HS cube (CCF) was developed for an additive Gaussian noise [10,11,12]. It processes all slices of the whole HS cube jointly, significantly improving the noise suppression and retrieving HS data that were lost due to noise. It is demonstrated in [10] that CCF algorithm produces good quality HS object reconstruction from extremely noisy data with SNR down to -8 dB. A great advantage of the CCF algorithm is that it is applicable for any type of complex-domain HS data, regardless of how it was obtained.

The contribution of this paper concerns an application of CCF filtering for HS object recognition and classification. We show that with the proper noise suppression by CCF, it is possible to significantly improve HSDH processing and to overcome problems of signal masking by background noise despite a high level of noise.

## 2. Problem Formulation

Let $U\left(x,y,\lambda \right)\subset C^{N\times M}$ be a slice of the complex-valued HS cube of the size N×M on (x,y) provided a fixed wavelength λ, and $Q_{\Lambda }\left(x,y\right)=\{U\left(x,y,\lambda \right),$
λ⊂Λ}, $Q_{\Lambda }\subset C^{N\times M\times L_{\Lambda }}$, be a whole cube composed of a set of wavelengths Λ with the number of individual wavelengths $L_{\Lambda }$. Thus, the total size of the cube is $N\times M\times L_{\Lambda }$ pixels.

The observations of the hyperspectral denoising problem under the additive noise assumption can be written as:(1)$$Z_{\Lambda }\left(x,y\right)=Q_{\Lambda }\left(x,y\right)+\varepsilon_{\Lambda }\left(x,y\right),$$ where $Z_{\Lambda },Q_{\Lambda }$, $\varepsilon_{\Lambda }\subset C^{N\times M\times L_{\Lambda }}$ represent the recorded noisy HS data, clean HS data and additive noise, respectively. Accordingly to the notation for the clean image, the noisy cube can be represented as $Z_{\Lambda }\left(x,y\right)=\{Z\left(x,y,\lambda \right),$
λ∈Λ}, $Z_{\Lambda }\subset C^{N\times M\times L_{\Lambda }}$ with the slices Z(x,y,λ).

The denoising problem is formulated as a reconstruction of unknown $Q_{\Lambda }\left(x,y\right)$ from the given $Z_{\Lambda }\left(x,y\right)$. The properties of the clean HS $Q_{\Lambda }\left(x,y\right)$ and the noise $\varepsilon_{\Lambda }\left(x,y\right)$ are essential for the algorithm development.

The denoising algorithm relies upon three basic and essential assumptions [13]: *Similarity* of the HS slices U(x,y,λ) for close values of λ follows from the fact that U(x,y,λ) are slowly varying functions of λ; *Sparsity* of the HS images U(x,y,λ) as functions of (x,y) means that there are bases such that U(x,y,λ) can be represented with a small number of elements of these bases; *Noise*
$\varepsilon_{\Lambda }\left(x,y\right)$ is zero mean Gaussian with unknown correlation matrix $L_{\Lambda }\times L_{\Lambda }$. Clean image subspace identification is a crucial first step in the developed algorithm. The signal and noise correlation matrices are estimated and then used to select the subset of eigenvectors that best represents the signal subspace in the least mean squared error sense.

### Complex Cube Filter Algorithm for HSI Denoising

Here we briefly describe Complex Cube Filter (CCF) algorithm (for more detailed description please see the paper [10]). The algorithm is used with the following notations:(2)$${\hat{U}}_{\bar{\Lambda }}\left(x,y\right)=\mathrm{CCF}\left\{Z_{\bar{\Lambda }}\left(x,y\right),\;\bar{\Lambda }\subset \Lambda \right\}.$$
Here, $\bar{\Lambda }$ is a set of slices to be denoised. CCF processes data of the noisy cube $Z_{\bar{\Lambda }}\left(x,y\right)$ jointly and provides the estimates ${\hat{U}}_{\bar{\Lambda }}\left(x,y\right)$ for all λ∈
$\bar{\Lambda }$. A principal flow work of the CCF algorithm is presented in Figure 1 and is composed of the following steps.
Calculation of the orthonormal transform matrix $E\subset C^{L_{\bar{\Lambda }}\times p}$ and the 2D transform domain eigenimage $Z_{2D,eigen}$ as:
(3)$$\left[E,Z_{2D,eigen},p\right]=HySime\left(Z\right),$$ where *HySime* stays for Hyperspectral signal Subspace Identification by Minimum Error [14] based on Singular Value Decomposition (SVD). *p* is a length of the eigenspace, which is automatically identified by a selection of the subset of eigenvalues that best represents the signal subspace in the least squared error sense. When *E* is given, the eigenimage is calculated as:
(4)$$Z_{2D,eigen}=E^{H}Z.$$Filtering of each of the N×M 2D images (slices) of $Z_{3D,eigen}$ by Complex-Domain Block-Matching 3D (CDBM3D) algorithm [15]:
(5)$${\hat{Z}}_{3D,eigen}\left(x,y,\lambda_{s}\right)=CDBM3D\left(Z_{3D,eigen}\left(x,y,\lambda_{s}\right)\right).$$ where *p* eigenvalues $\lambda_{s}$ belong to the eigenspace.Returning from the eigenimages of the transform domain to the 2D original image space as follows
(6)$${\hat{Z}}_{2D}=E{\hat{Z}}_{2D,eigen}.$$

Forward and backward passages 2D⇆3D signed between the algorithm steps define the reshapes from 2D to 3D and vice versa for proper processing of the HS data, the subscript (2D or 3D) denotes the number of dimensions. It is needed to produce SVD transform since it works only with 2D data, and to produce the CDBM3D filtering in the corresponding 3D domain $Z_{3D,eigen}$ slice-by-slice. However, in order to return these filtered data ${\hat{Z}}_{3D,eigen}$ into the original image space we need to use 2D transform (Equation 6) and, thus again to reshape data into 2D transform space. Optimization in step 1 of the HS data results in minimization of its size and usually obtained subspace dimension much smaller than the initial one $p\ll L_{\bar{\Lambda }}$. This leads to a shorter processing time of the algorithm. Thus, the CDBM3D filtering is produced only for *p* eigenimages but the backward transform (Equation 6) gives the estimates for all $L_{\bar{\Lambda }}$ spectral images.

## 3. Spectral Analysis

Traditionally, for spectral analysis of an object, unknown spectra obtained in the experiments are compared with the known standard spectral fingerprint databases [16]. Otherwise, standard spectral fingerprints might be obtained directly from the experimental HS cube. For this purpose, averaging of spectra from an area with known objects’ material (or process) is produced. Such fingerprint creation is free from external databases and, therefore, is useful for the new processes or object detection which are not recognized before and do not have standard databases. However, this approach is sensitive to noise in data and, thus, a recognition becomes complicated and might even fail.

A comparison of the fingerprint spectrum with the spectrum in question can be produced based on the correlation analysis. In this paper, for this analysis we use the formula:(7)$$\rho \left(A,B\right)=\frac{1}{N-1}\sum_{i=1}^{N}\left(\frac{A_{i}-\mu_{A}}{\sigma_{A}}\right)\left(\frac{B_{i}-\mu_{B}}{\sigma_{B}}\right).$$

Here *A* and *B* are two compared spectra; $\mu_{A}$ and $\sigma_{A}$ are the mean and standard deviation of a spectrum A, respectively, and $\mu_{B}$ and $\sigma_{B}$ are the mean and standard deviation of fingerprint spectrum B; N is a number of points (spectral lines) in the spectrum. In this comparison, the averaging is produced over the whole spectral interval and it should be done for all pixels of the images in a pixel-by-pixel manner. Thus, as a result, we obtain an image of the correlations ρ(A,B).

## 4. Spectral Object Recognition

For objects recognition demonstration, we took experimental HS data obtained by transmissive HS digital holography technique [7], we do not give a detailed description of the HSDH technique here to emphasize that CCF algorithm does not depend on the way how HS data is obtained. For the given data the lowest SNR =6.7 dB is for the slice λ=787 nm, the highest SNR =20.5 dB is for the slice λ=576 nm, and average SNR =12.6 dB. The investigated object was a transparent sample with different color letters, which correspond to different spectra. The object slice is demonstrated in Figure 2 and the used light source spectrum in Figure 3. As it is seen from Figure 2, the investigated object has a simple structure with 3 colors (red, green, blue) and the transparent area which should be recognized. In the right image of Figure 2, the small squares show areas of the object of different colors selected in order to define the standard fingerprints. An HS cube corresponding to that object consists of 807×807×226 pixels, which in turn corresponds to 651249 spectra to be recognized.

Using CCF algorithm, we produced a noise suppression for the initial noisy HS cube (left column) and obtain filtered HS cube (see Figure 4, middle column). The top row of Figure 4 demonstrates amplitudes and the bottom one - phases. The HS slice in Figure 4 corresponds to 599 nm and illustrates effective noise suppression and object details recovery from the noisy slice by the CCF algorithm. To emphasis spectral recognition improvement, we compare CCF filtering results not only with initial noisy data but also with the Fast Hyperspectral Denoising (FastHyDe) technique [13], which is developed for HS imaging but only for real-valued data. The filtering result by FastHyDe is in the right column of Figure 4, it produces also visually good noise suppression for the amplitudes, but do not provide correct phase filtering.

In Figure 5, the normalized spectrum fingerprints for each color of the object are presented, they are obtained from noisy (solid lines with circles), filtered by CCF (dash lines with triangles), and filtered by FastHyDe (dash-dot lines with squares) HS data. The fingerprints for each color have distinctive peaks in the corresponding range of the wavelengths, however, it is clearly seen that for the fingerprints obtained from the noisy HS data background signal is higher, especially for the red and blue spectra. Due to this noisy background, the true signal information might be lost. For the FastHyDe spectra, the extra peaks are registered for red, green, and blue colors, which might be explained by a leakage of the signal during the SVD transform. For CCF fingerprints we do not observe such behavior due to effective noise suppression which along with the SVD transform utilizes amplitude-phase correlations in CDBM3D.

Figure 6 demonstrates correlation maps (images) obtained by (Equation 7): left column is for noisy HS data, central column is for filtered by CCF, and the right one is for the filtered by FastHyDe. Colorbar relates to all images and represents correlation value, the white color corresponds to high correlation, black to the low one. Even for such a simple object, it is hard to recognize different spectral regions due to the high noise level in the HS data. From noisy data, object recognition in the blue and red regions is failed since the noise level in these spectrum regions was high compared to the level of intensity of the used LED masking the true signal. Green and white regions are recognized better but with low correlation values with the standard fingerprints.

For FastHyDe correlation maps, we observe higher correlation values for the whole object with barely separated object features. It is explained by the similarity of the fingerprints provided by FastHyDe (see Figure 5). Completely different correlation maps are for CCF filtered HS data, where all different colors are well recognized. For the red, green, and blue colors of the object, the recognition improvement due to CCF filtering is obvious since the corresponding features have the highest correlations.

To evaluate the performance of the recognition, in Figure 7 we demonstrate Receiver Operating Characteristic (ROC) [17] curves for recognition from each data set for every color of the object with different correlation thresholds from 0 to 1, step 0.1. To estimate a recognition probability, logistic regression was used [18]. In Figure 7, solid curves with circles correspond to recognition from noisy HS data, dash curves with triangles correspond to recognition from filtered by CCF HS data, and dash-dot curves with squares—to filtered by FastHyDe. According to ROC curves, the best recognition is performed for blue color features from filtered by CCF HS data, the second-best is for recognition from green color also filtered by CCF HS data. Significant recognition improvement comparatively to noisy data is demonstrated for the red color, where for noisy data it is almost a random guess. In the region of low false-positive rate values, the highest true positive rate values for each color except white are also for the CCF filtered data. For the white color, no improvement is obtained which is explained by the good-enough fingerprint spectrum obtained from the noisy data. FastHyDe filtering has not improved recognition for white, blue, and green colors, however, it significantly improved red color recognition, which is supported also by a high value of area under ROC curves (AUC), presented in Table 1. The highest AUC values are for filtered data by CCF for green and blue colors, which confirms detection improvement due to CCF noise suppression. For the white region, AUC is higher for noisy data, which means that for the transparent regions of the object (places without features) there is no need in noise suppression since, for the given spectral analysis, the fingerprint spectrum correlation with the estimated ones is high.

## 5. Conclusions

Application of the complex domain HS cube filter to the HS object recognition is performed. It is demonstrated that, with a proper noise suppression, it is possible to significantly improve the classification process. It becomes possible to recognize spectral features masked by a noise. We would like to emphasize that due to general denoising problem formulation CCF algorithm works with any type of HS data with additive Gaussian noise, therefore it is applicable for a wide range of tasks. Due to such flexibility, it is compatible with other more sophisticated recognition techniques, as for example with the use of convolutional neural networks [19].

## Figures and Tables

**Figure 1 sensors-19-05188-f001:**
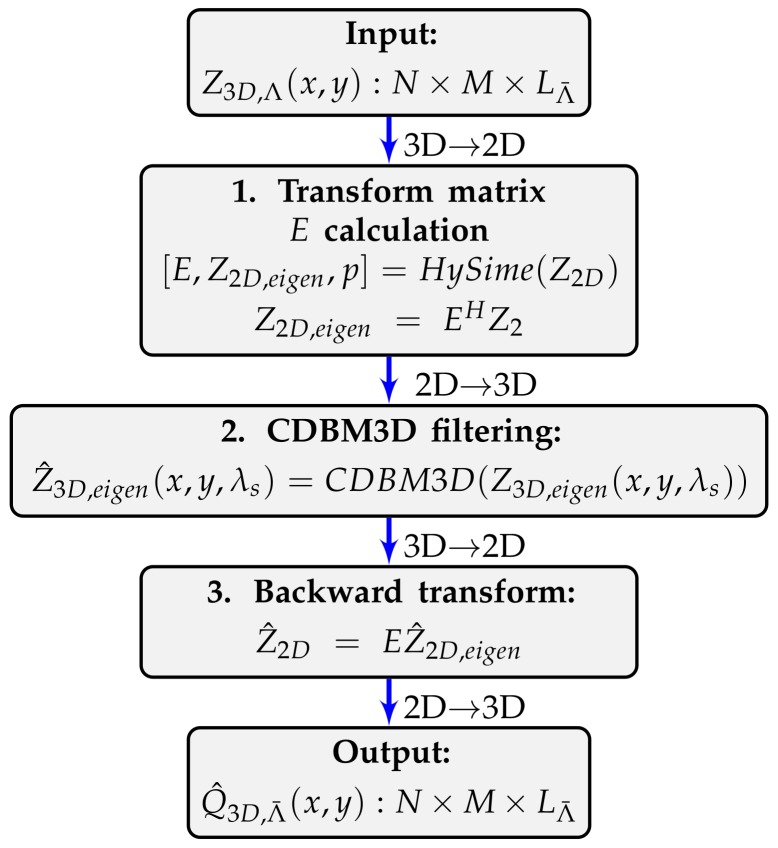
Complex Cube Filter (CCF) algorithm. Data arrays reshaping from 2D to 3D and vice versa are depicted by arrows 2D⇆3D.

**Figure 2 sensors-19-05188-f002:**
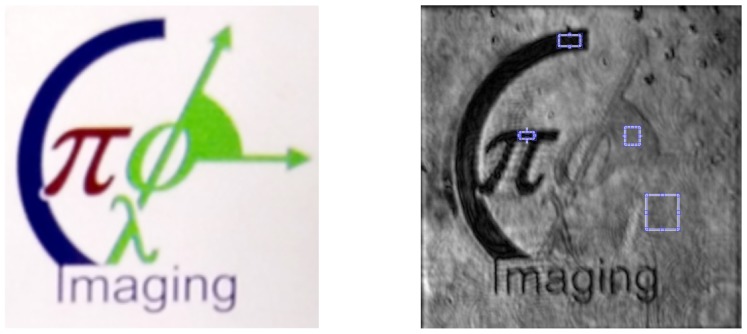
Left image, the used transparent object slide, the white color corresponds to the transparent area. Right image, a slice of the HS cube with the small squares corresponding to different colors of the object. The standard fingerprints are produced by taking spectra from these squares.

**Figure 3 sensors-19-05188-f003:**
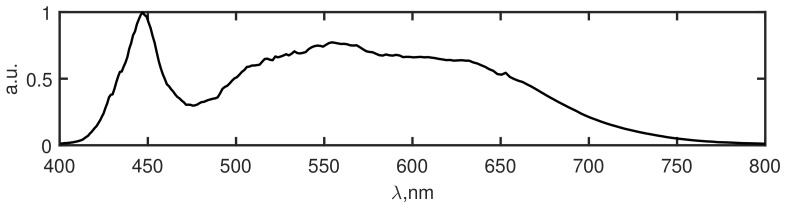
Used LED spectrum.

**Figure 4 sensors-19-05188-f004:**
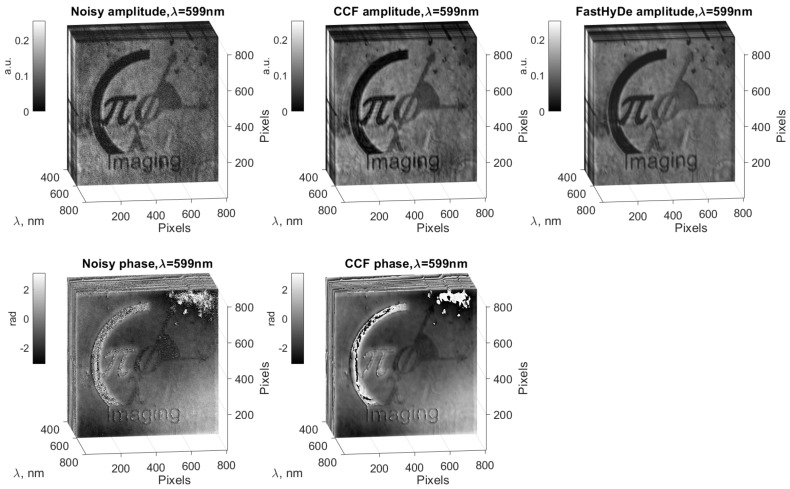
Noisy (left column) hyperspectral imaging (HS) cube and filtered HS cubes by CCF (center column) and FastHyDe (right column). The front slice corresponds to 599 nm, SNR =14.6 dB. The top row is for amplitude images and the bottom row is for phases. FastHyDe is developed for real data processing and does not provide phase filtering.

**Figure 5 sensors-19-05188-f005:**
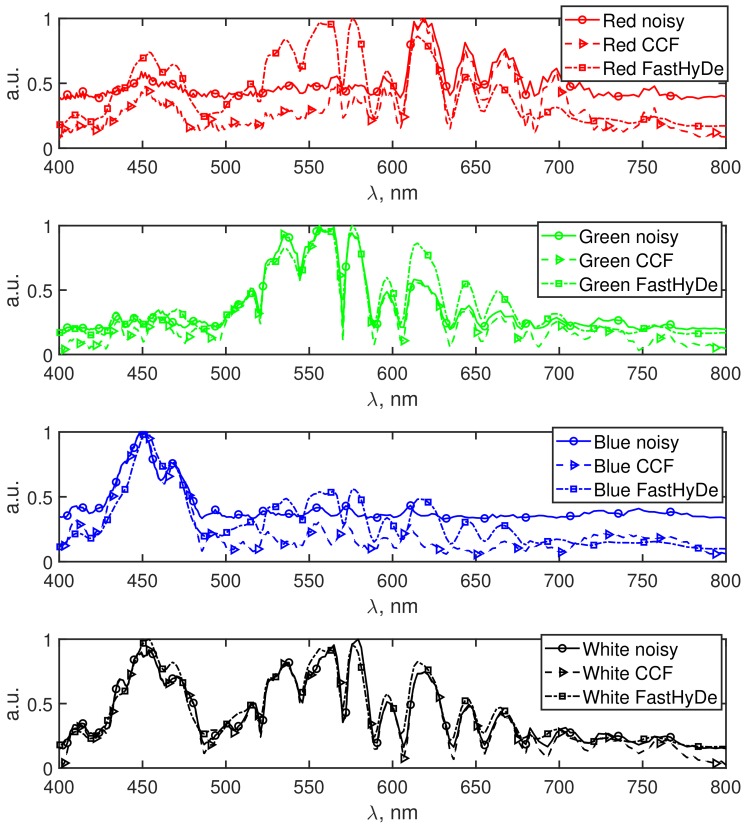
The fingerprint spectra for the red, green, blue, and transparent (white) areas obtained in the small squares in Figure 2 before (solid lines with circles) and after filtering by CCF(dash with triangles) and FastHyDe (dash-dot with squares). Curve colors correspond to the color of the object features.

**Figure 6 sensors-19-05188-f006:**
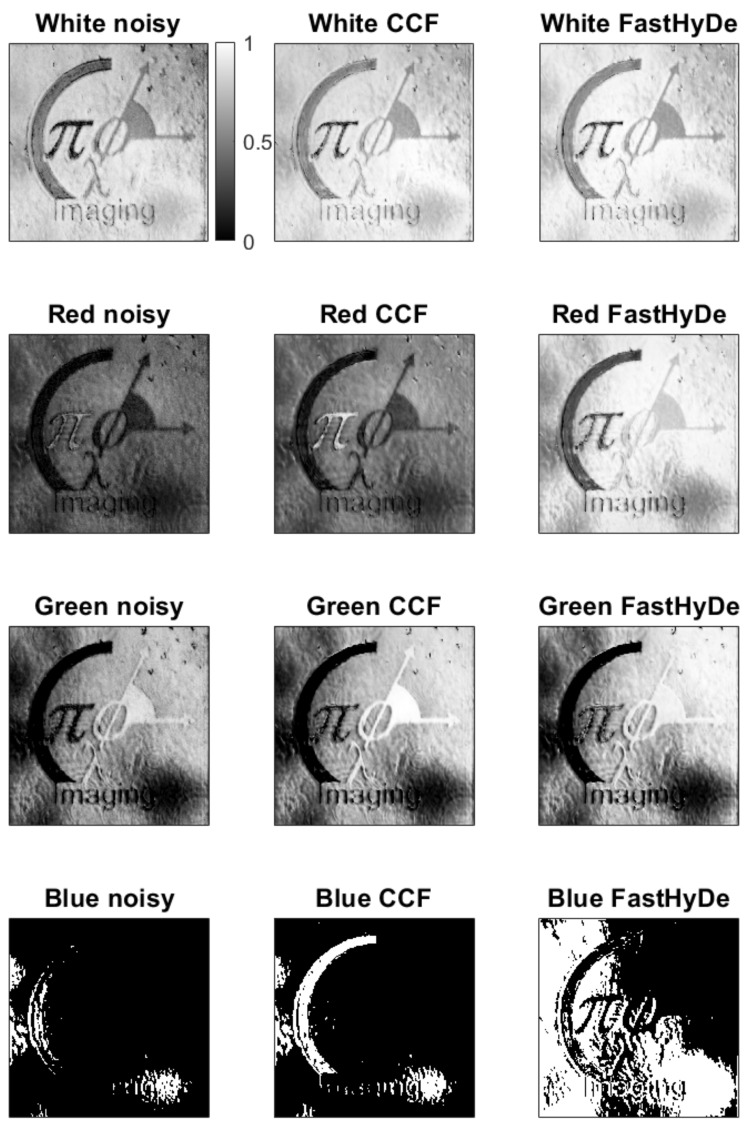
Correlation maps. The left column is for noisy HS data, the central one is for filtered by CCF, and the right one is for the filtered by FastHyDe. From top to bottom are spectral colors: white, red, green, blue. White color in maps corresponds to the higher correlation and recognized pixel for a given color.

**Figure 7 sensors-19-05188-f007:**
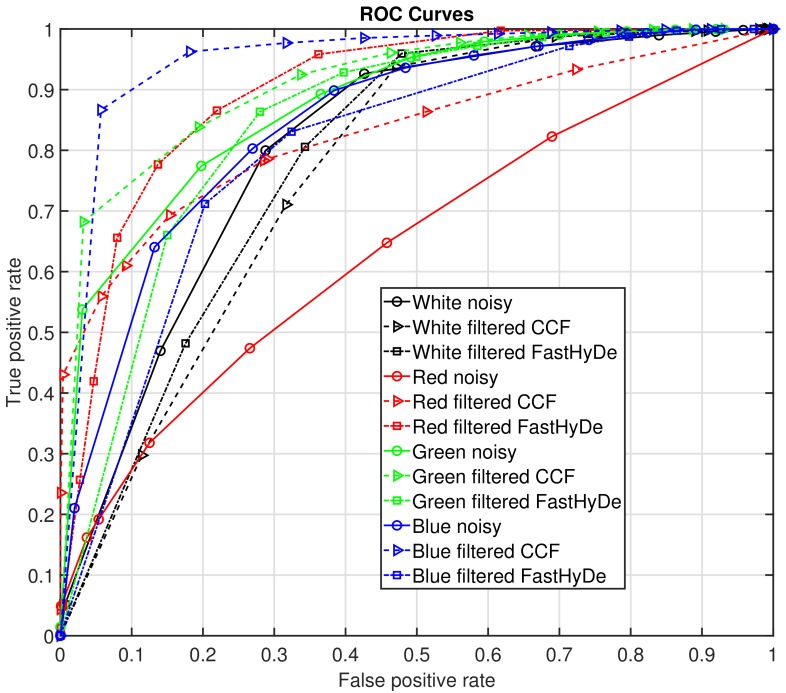
Receiver operating characteristic curves for recognition of different colors in HS data: noisy (solid curves with circles), CCF filtered (dash curves with triangles), and FastHyDe filtered (dash-dot curves with squares). Curves’ colors correspond to the color of the object features.

**Table 1 sensors-19-05188-t001:** AUC.

Color	Noisy Data	CCF	FastHyDe
red	0.6416	0.8293	0.8986
green	0.8767	0.9102	0.8443
blue	0.8428	0.9478	0.8008
white	0.8054	0.7652	0.7838

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
