# Peer review of "Spectral Object Recognition in Hyperspectral Holography with Complex-Domain Denoising"

_sensors, 2019, doi:10.3390/s19235188_

Round 1

Reviewer 1 Report

This paper applied an application of Complex Cube Filter (CCF) to suppress noise for HS object classification and recognition. With the proper noise suppression, it is possible to significantly improve the classification results and recognize spectral features which are not recognized in noisy data. It has some merits, but also with several problems. Therefore, I suggest to major revise this paper with following comments in details.

ABSTRACT

Abstract is well written and easy to follow.

Introduction

It is well written and serves the script's research topic; The current research status of noise reduction methods is not sufficiently described. In the introduction, this paper indicated that with proper noise suppression by CCF, it is possible to significantly improve HIS processing and to overcome problems of signal masking by background noise despite a high level of noise. How to define the noise of high level. Can any types of noise in HSI be suppressed?Please give applicable range of noise.

Problem formulation

In the first paragraph, it is explained that Q is a complex function, i.e. the initial hyperspectral data, the non-denoised hyperspectral data, which is unreasonable with the symbol definition in formula (1) below; There are few innovations in the theoretical part: The concrete formulas are almost from references. Where are the new theoretical formulas put forward in this paper. The CCF algorithm can be summarized into three steps. The first is calculating the transform matrix E to obtain the signal eigenimage Z. The second is applying the CDBM3D filter to the signal eigenimage Z. The last is backward transforming to original domain. When the method transforms the original HSI to obtain the signal image Z, how to set the parameter p? In Figure 1, the sign is not clearly defined.

Spectral analysis

It is well written and easy to follow.

Spectral object recognition

The experimental data are insufficient: In the experimental results, there is no comparative experiment to highlight the advantages of complex domain denoising and to show the effect of noise suppression on target recognition. This paper neither compares the experiment with the original hyperspectral image which without denoising, nor compares with the traditional denoising methods. In Figure 5, the comparation between noisy spectra and filter spectra cannot prove the filter spectra can keep the true signal information, the authors should add the standard spectra of four objects to demonstrate that the proposed algorithm can recover the spectra of signal more accuracy. In figure 6, using a fixed threshold to obtain the binary maps to evaluate the performance of detection is not suitable. Please use ROC to evaluate the performance of detection with different threshold from 0 to 1. According to the article, if CFF can improve noise suppression and retrieves HS data, please draw the ROC and calculate the AUC value using the detection rate and false alarm rate to show the performance of noise suppression and signal recovery which the method improves.

Conclusion

The conclusion shows that the method proposed in this paper can be applied to any model, but it is not reflected in part V experiments.

Reviewer 2 Report

This paper is an application of the Complex Cube Filter algorithm for spectral object recognition. While I do not see any reason why this paper should not be published I would like to see a more interesting experiment in recognition in addition. Like the detection of real objects like trees, grass, water etc. 

Author Response

This paper is an application of the Complex Cube Filter algorithm for spectral object recognition. While I do not see any reason why this paper should not be published. 

Response: We have made English language corrections of the manuscript.

I would like to see a more interesting experiment in recognition in addition. Like the detection of real objects like trees, grass, water etc

Response: There are databases for real-valued objects as for example (AVIRIS database), but there are no existing complex-valued databases.

Round 2

Reviewer 1 Report

Thanks for the authors who took time to address my comments. I have no more comments except for the followings:

(1) Reference is missed in P3L71-72,"where HySime stays for Hyperspectral signal Subspace Identification by Minimum Error [?
]";
(2) Explanation in P5L109-110 is not consistent with Figure 4.

Author Response

We are thankful for the reviewer's comments:

(1) Reference is missed in P3L71-72,"where HySime stays for Hyperspectral signal Subspace Identification by Minimum Error [?]";

It is corrected.

(2) Explanation in P5L109-110 is not consistent with Figure 4.

It is corrected.

And also we have provided English editing of the manuscript.  

Reviewer 2 Report

Fine by me. 

Author Response

We are thankful for the reviewer's comments and provided English editing of the manuscript.